

# Prevalence and risk of spinal pain among physiotherapists in Poland

Sebastian Glowinski[1,2,*], Aleksandra Bryndal[2,*] and Agnieszka Grochulska[2]

[1] Department of Mechanical Engineering/Division of Mechatronics and Automatics, Technical University of Koszalin, Koszalin, Zachodniopomorskie, Poland

[2] Institute of Health Sciences, Slupsk Pomeranian Academy, Slupsk, Pomorskie, Poland

[*] These authors contributed equally to this work.

## ABSTRACT

**Background**. The purpose of this study was to determine the prevalence, symptoms of, and risk factors for spinal pain in physiotherapists, as well as to analyse the correlation between these factors and the nature of the work, anthropometric features of the respondents, and the level of their physical activity.

**Methods**. The study was conducted among 240 physiotherapists (71 male and 169 female) with a mean age of 38.7 years. They were divided into three groups: physical therapy (37), kinesitherapy (158) and massage (45). Physiotherapists were evaluated with a specially designed questionnaire, the postural discomfort chart, the Neck Disability Index (NDI) questionnaire, and the Oswestry Disability Index (ODI) questionnaire.

**Results**. The analysis showed a 91.7% incidence of spinal pain in physiotherapists (91.1% for kinesitherapy, 97.3% for physical therapy, and 88.9% for massage). The study revealed that 50.2% of physiotherapists indicated one to five pain episodes in their careers. Most respondents reported pain in the lumbosacral spine (82%) and the cervical spine (67%). Most respondents (58.5%) scored the pain as moderate (VAS scale). Carrying (62.6%) and torso bending (37.4%) were indicated as the causes of pain.

**Conclusions**. Physiotherapists demonstrate a high prevalence of spinal pain. Physical activity reduces the frequency of pain episodes.

## INTRODUCTION

Neck pain (NP) and low back pain (LBP) are very common musculoskeletal disorders and leading causes of disability worldwide (*Vassilaki & Hurwitz, 2014*). LBP is a complex therapeutic and diagnostic problem. Numerous epidemiological data show that 11%–84% of people experience, have experienced, or will experience pain in the lower spine at least once in their lives (*Hoy et al., 2012*; *Walker, 2000*; *Hoy et al., 2010*). With a global prevalence of 9.4%, LBP was ranked highest in the number of years lived with a disability and sixth in overall burden of disease (*Hoy et al., 2010*). The nature of occupational work related to musculoskeletal load promotes pain in the lower spine (*Steenstra et al., 2017*;

Corresponding author
Sebastian Glowinski,
sebastian.glowinski@tu.koszalin.pl

*Glowinski et al., 2020*). Pain very often occurs during lifting weights, frequently repeated bending over and rotations in the lower part of the spine, and maintaining one forced body position for a long time (*Vicente-Herrero & Tulio, 2019*; *Marras, 2000*). NP is also very common (*Hoy et al., 2012*; *Hoy et al., 2010*), with a general prevalence in the general population of 0.4%–86.8%. With a global prevalence of 4.9%, NP ranks fourth in terms of general disability and 21st in terms of the overall burden of disease (*Hoy et al., 2010*). LBP is the most common musculoskeletal occupational disorder (*Marras, 2000*). Worldwide, it is estimated that 37% of LBP cases are work-related, and this problem causes 818,000 cases of disability per year (*Punnett et al., 2005*). As such, LBP is an economically important problem in developed countries. There is growing evidence that NP is associated with many occupational factors, including physical work requirements and work-related psychosocial and organizational factors. In particular, studies have shown that abnormal postures, strenuous physical work, and repetitive and precise work are physical risk factors for NP. The occurrence of pain in the cervical part of the spine impacts the economic situation related to productivity, workers' compensation, as well as the reduction of the number of workers and their well-being throughout their lives (*Yang et al., 2016*; *Bryndal et al., 2019*).

The causes of NP and LBP are multi-factorial, with age, sex, genetic makeup, obesity, environment and occupation playing significant roles (*Williams & Sambrook, 2011*; *Heuch et al., 2010*; *Bryndal et al., 2020*). Occupational factors associated with NP and LBP include the fast pace of work, repetitive movement patterns, insufficient recuperation time, heavy lifting, other strenuous manual work, non-neutral body postures, mechanical pressures, bending, twisting, vibrations and low temperature (*Punnett et al., 2005*; *Adams, 2013* ).

In the profession of a physiotherapist, the main occupational risk factors affecting the locomotor system are dynamic physical load (physical effort, monotype of movements), static physical load (forced body position), and the possibility of falling. In general, the professional activities performed by PT are mainly physical work in combined planes. The way they are performed determines the forced body position. Movements within the same joints and forced positions are usually performed in a short time, but many times during the day, which leads to significant strain on the locomotor system. This results in a particularly high risk of injury or overload of spinal muscles, spinal ligaments or, ultimately, intervertebral discs (*Milhem et al., 2016*; *Campo et al., 2008a*).

Physiotherapy (PT) comprises three main sections: kinesitherapy (movement therapy, therapeutic gymnastics), physical therapy (treatment involving physical stimulation, either natural or produced by devices), and massage. In each of these departments, the PT has a different nature of work, which may cause different overloads in the locomotor system. The main threats that are specific to a given specialty of physiotherapy include a mechanical overload of the musculoskeletal system (lifting patients, equipment, frequent repetition of the same movements); forced body position; bending and rotation of the torso with load; insufficient equipment for lifting and transferring patients; improper habits of lifting and carrying patients; unpredictable patient movements or falls. For the physical therapy specialty, it is routine and repetition of selected activities; the one-sided load on the musculoskeletal system; forced body position during the activity. For specialties, massage is a long-term leaning during the activity; mainly standing work; static physical workload

with torso flexion and rotation; dynamic physical load - physical effort, monotype of movements (*Milhem et al., 2016*; *Campo et al., 2008a*).

Nowadays, the development of pathologies within the spine is strongly associated with the economic situation, and the working time of physiotherapists is significantly longer than the statutory limit, often reaching ten hours a day. For spinal pain to develop, many other factors must also occur. Among the most important ones are the anthropometric features of a given person, the level of his or her physical activity (since the physical nature of the work should not be identified with physical activity), genetic determinants, and history of traumas or diseases that may affect spinal function. The risk of disorders of spinal functions manifested by spinal pain is likely to be higher with the simultaneous impact of several factors mentioned above (*Macintosh & Bogduk, 1991*; *Williams, 1955*). Taking into account the high frequency of spine problems among physiotherapists, the question arises whether the specificity of work in the selected specialization of physiotherapy is associated with a higher risk of these problems. So far, in the literature, in the assessment of factors contributing to the formation of back pain in the profession of a physiotherapist, no division into specialties has been applied due to the nature of the work in which a given physiotherapist works the most time (kinesitherapy, physical therapy, massage). We took this division in our manuscript.

This study aims to analyse the incidence of spinal pain in Polish physiotherapists divided into specialties (kinesitherapy, physical therapy, massage), with consideration of potential risk factors related to their occupation, anthropometric features of respondents, and the level of their physical activity.

Specific objectives:

- To identify the incidence of lumbar and cervical spine pain in physiotherapists.
- To analyze the significance of this medical problem in quantitative terms.
- To describe the most common body positions and activities related to the nature of work causing pain in the examined occupational group.
- To analyze a possible relationship between the occurrence of lumbar and/or cervical spine pain and the nature of the work.
- To determine the degree of disability caused by spinal pain in physiotherapists.
- To assess the influence of physical activity on the occurrence of pain.

The results are to be used in the future to design an exoskeleton supporting physiotherapists at work.

## MATERIALS & METHODS

A total of 240 people (169 females (70.4%) and 71 males (29.6%)) licensed to practice as a PT in Poland completed an anonymous questionnaire focused on spinal pain. The study protocol was approved by the Bioethics Committee at the district medical chamber in Gdansk (KB-14/20). The informed consent form was written at the beginning of the test. It was minimizing the possibility of coercion or undue influence, and the subject had sufficient time to consider participation. Research-related information were presented to enable people to voluntarily decide whether or not to participate as a research subject. We

advised that the results would be used in medical research and suggested that the answers be honest. An electronic questionnaire was sent to Polish public and non-public medical facilities employing physiotherapists, and to private physiotherapy practices. Participation in the study was declared by 286 physiotherapists. Of these, 16 met the exclusion criteria, and another 30 did not complete the survey. Subjects who were older than 18 years of age, had a valid licence to practice their profession, and actually worked as a physiotherapist were included. Subjects younger than 18 years of age, those with a history of spinal injury, history of spinal surgery, deformities in the spine and/or lower limbs, and pregnant women (due to potential pregnancy-related spinal pain) were excluded from the study. The data were collected using questionnaires: a questionnaire designed by the authors of this study, the Polish version of the Neck Disability Index (NDI) (*Misterska, Jankowski & Glowacki, 2011a*), and the Polish version of the Oswestry Disability Index (ODI) (*Misterska, Jankowski & Glowacki, 2011b*).

In the questionnaire designed by the authors the respondent defined or described the characteristics of his/her pain. The questions focused on the location of spinal pain (in the cervical, thoracic or lumbar segment), its duration, persistence of symptoms, suspected cause(s), and the reasons for the severity of these ailments.

The nature of the work was described in terms of its duration, the type of activities prevailing during the work, the number of working hours, and the number of years effectively worked. Work done by physiotherapists was categorized into three specialities: kinesiotherapy (movement therapy, therapeutic gymnastics), physical therapy (treatment involving physical stimulation, either natural or produced by devices), and massage. The level of physical activity among respondents (broadly defined recreation) was also determined. Age, body weight and height values of participants were recorded (BMI was calculated). Body mass index (BMI) was calculated as weight (kg) divided by height squared ($m^2$).

Oswestry is a common condition-specific tool that has been used in over 200 published articles since its inception in 1980 (*Vassilaki & Hurwitz, 2014*; *Hoy et al., 2012*). The Polish version of the Neck Disability Index (NDI) was used to assess cervical pain (*Thomas, Walsh et al., 2003*). It consists of 10 questions concerning: pain intensity, care, lifting objects, reading, headache, ability to focus, work, driving, sleeping, and rest. The Polish version of the Oswestry Low Back Pain Disability Index (ODI) was used to assess disability caused by lumbar pain (*Misterska, Jankowski & Glowacki, 2011b*). It contains 10 questions concerning: pain intensity, care, lifting, walking, sitting, standing, sleeping, sexual life, social life, and travelling. In both questionnaires, each question is scored from 0 to 5. All scores are summed and divided into the highest possible score of 45, producing a 0 to 100-percentage scale, with 0 representing no disability and 100 representing complete disability. For missing responses, the total possible score is reduced (e.g., the highest possible score for eight responses would be 40). For ease in clinical interpretation, this score was then subtracted from 100. Thus, 0 represents complete disability and 100 represents a normal function. The inversion of the score does not affect any statistical calculations or mathematical relationships.

The aggregate NDI and ODI were presented as a score in the range from 0 to 50 or percentages of 0–100%. Score 0–4 (0–8%) indicated no disability, score 5–14 (10–28%) indicated minimal disability, score 15–24 (30–48%) indicated moderate disability, score 25–34 (50–64%) indicated severe disability, score 35–50 (70–100%) indicated extreme suffering and disability.

Responsiveness is only one characteristic to consider when choosing a survey instrument. For example, *Walsh et al., (2003)* suggested that the ODI is shorter than the SF-36, and if computerized survey administration is not available, it may be easier to administer and score. If ease of administration and scoring is a top priority, then one may choose to utilize the ODI, and responsiveness would not significantly suffer (*Walsh et al., 2003*). The cost of such a decision at least includes the loss of general health information, the ability to compare disability between patients with differing conditions, and the possible identification of unintended side effects from the new treatment. The concept of responsiveness is challenging. *Beaton (2000, 2001)* have suggested that the responsiveness of an instrument should be viewed within the context of "who" is being studied, "which" scores are being contrasted, and "what" type of change is being assessed. The issue of which condition-specific survey is "best" has been studied by *Leclaire et al., (1997)*. The authors prove that the ODI scale measures functional ability. Moreover, according to the authors, the ODI appeared more sensitive in the severely disabled patients than, for example, the Rolland Morris disability scale. ODI may be more suitable for patients with greater limitations. NSN has chosen to utilize the ODI over the RMQ. Our previous work suggests the ODI would be the best measure given our patient population (*Bryndal et al., 2020*; *Glowinski & Krzyzynski, 2013*). Truthfully, there is no gold standard for measuring change. We agree with the reviewer that questions used to assess spine pain are somewhat generally defined. The questions did not specify a spine problem but did refer to the patient's specific musculoskeletal condition.

All statistical calculations were performed using STATISTICA software, version 13.3. (StatSoft, 2020, http://www.statsoft.com). Quantitative variables were characterized by the mean, standard deviation, median, minimum and maximum values (range) and 95% CI (confidence interval). Qualitative variables were presented as numbers and percentage values (percentage). Quantitative variables were tested for the normality of distribution using different tests: W Shapiro–Wilk, Lilliefors, Kolmogorov–Smirnov and Jarque–Bera. The Statistica package automatically suggests the four types of tests for the normality of distribution. The Brown-Forsythe test was used to verify the hypothesis on the equality of variances due to different numbers of variables (disciplines of PT).

## RESULTS

The PT group consisted of professionals mainly providing kinesitherapy, physical therapy or massage as part of their work. Basic quantitative data are presented in Table 1.

The mean BMI of all PTs was 24.1 (3.4). The mean BMI in women was normal (22.6), while men were slightly overweight (26.0) (Fig. 1).

Non-radiating pain in the cervical spine was reported by 67 (29.8%) respondents, and pain radiating to one limb was reported by 36 respondents (16.0%). Radiation to both
**Table 1  Characteristics of the study group by speciality.**

| Variable | women<br>men | All groups<br>(N = 240)<br>169 (70.5%)<br>71 (29.5%) | Kinesitherapy<br>(n = 158)<br>109 (69.0%)<br>49 (31%) | Physical therapy<br>(n = 37)<br>28 (75.7%)<br>9 (24.3%) | Massage<br>(n− = 45)<br>32 (71.1%)<br>13 (28.9%) |
|---|---|---|---|---|---|
| | | mean (SD);<br>med; [min; max] | mean (SD);<br>med; [min; max] | mean (SD);<br>med; [min; max] | mean (SD);<br>med; [min; max] |
| Age [years] | | 38.7 (11.0);<br>35; [19; 63] | 39.1 (11.1);<br>36; [19; 63] | 32.8 (5.5);<br>34; [23; 47] | 42.2 (11.9);<br>38; [22; 61] |
| | women<br>men | 40.3 (11.4)<br>35.0 (8.9) | | | |
| Height [cm] | | 171.0 (8.6);<br>170; [153; 199] | 170.9 (8.7);<br>170; [153; 199] | 171.4 (8.0);<br>170; [159; 189] | 170.7 (8.7);<br>170; [155; 198] |
| | women<br>men | 167.1 (5.9)<br>180.3 (6.6) | | | |
| Weight [kg] | | 79.9 (13.9);<br>67; [42; 120] | 70.8 (13.7);<br>68; [43; 120] | 69.8 (14.6);<br>64; [52; 105] | 72.0 (14.5);<br>68; [42; 112] |
| | women<br>men | 64.5 (8.9)<br>86.1 (11.8) | | | |
| Employment [years] | | 15.1 (11.8);<br>12; [1; 42] | 15.3 (11.7);<br>12; [1; 40] | 8.4 (5.2);<br>9; [1; 18] | 19.7 (13.6);<br>15; [1; 42] |
| | women<br>men | 15.1 (11.8)<br>11.5 (8.8) | | | |
| Work time per day<br>less than 6<br>from 6 to 8<br>over 8 | | 46 (19.2%)<br>139 (57.9%)<br>55 (22.9%) | 30 (19.0%)<br>89 (56.3%)<br>39 (24.7%) | 9 (24.4%)<br>16 (43.2%)<br>12 (32.4%) | 7 (15.6%)<br>34 (75.6%)<br>4 (8.8%) |
| less than 6 | women<br>men | 36 (21.3%)<br>10 (14.1%) | | | |
| from 6 to 8 | women<br>men | 110 (56.1%)<br>29 (40.8%) | | | |
| over 8 | women<br>men | 23 (13.6%)<br>32 (45.1%) | | | |

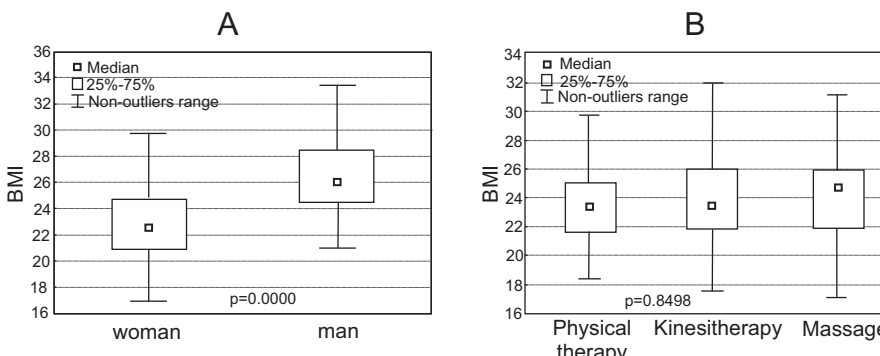

**Figure 1  Box plot representation of the distribution of physiotherapists' Body Mass Index (A) BMI vs Sex (B) BMI vs specialization (physical therapy, kinesitherapy, massage).**

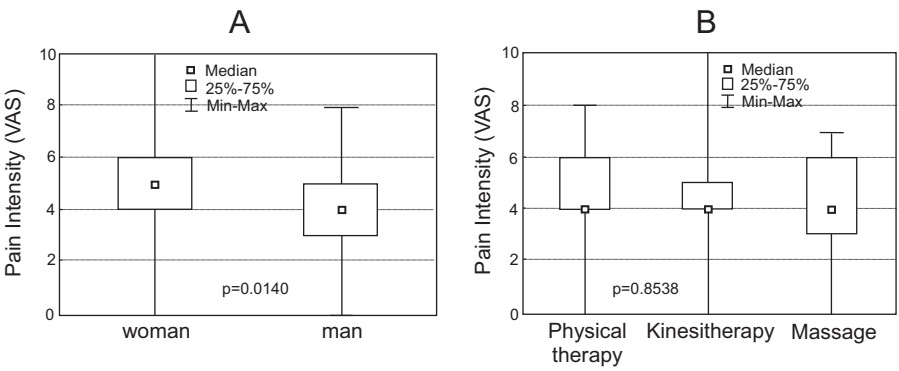

**Figure 2** **Box plot representation of pain intensity (A & B).**

extremities due to cervical pain was reported by 10 (4.4%) PTs. Non-radiating pain in the thoracic spine was reported by 56 (24.9%) respondents, while radiating pain was reported by 29 respondents (12.9%). Pain defined as non-radiating in the lumbar-sacral spine was reported by 110 (48.9%) respondents, radiation to one leg was reported by 83 respondents (36.9%), while radiation to both legs was reported by 6 respondents (2.7%).

Most PTs (125; 55.3%) scored the last pain episode between 4 (moderate) and 5 (moderate/severe; VAS scale). In the case of kinesitherapy, it was 82 (51.9%), physical therapy 21 (56.7%) and massage 21 (46.7%). Only 57 (25.3%) of all respondents declared more severe pain. The remaining 42 (17.9%) subjects declared very mild to mild/moderate pain. Pain intensity in particular groups is presented in Fig. 2.

No painkillers were used by 41.6% of respondents, 57.1% used medication when needed, and 1.3% used low doses on a regular basis. In 40.7% of respondents the pain did not reduce mobility, in 46.5% it partially restricted it, in 11.4% of respondents pain made work difficult, and in 0.9% of respondents pain prevented independent functioning. Detailed data by speciality are presented in Table 2.

In the case of the cause of pain and the activities that exacerbate it, the sum will not be 100% since respondents could give more than one answer. The data refer to the whole group.

The results of the ANOVA test indicated that a zero-hypothesis stating that the mean intensity of the last pain episode (VAS) was similar in individual groups can be accepted at 0.585. Tukey's post hoc test also confirmed this relationship. A graphical interpretation is presented in Fig. 3A. The points correspond to the mean values in the individual groups, and the error bars show confidence limits around the mean. According to the results obtained, the lowest point is the one corresponding to the intensity of the last pain episode among masseurs. The value 3 on the vertical scale indicates mild/moderate pain, while 5 indicates moderate/severe pain. In the case of reduced physical activity, a zero hypothesis can also be accepted for individual groups at 0.770. Figure 3B shows the relationship. On

**Table 2  Characteristics of the group with a breakdown by speciality in terms of the occurrence of pain.**

| Variable | | All groups (N = 240) | Kinesitherapy (n = 158) | Physical therapy (n = 37) | Massage (n− = 45) |
|---|---|---|---|---|---|
| **Pain** | | | | | |
| yes | | 220 (91.7%) | 144 (91.1%) | 36 (97.3%) | 40 (88.9%) |
| no | | 20 (8.3%) | 14 (8.9%) | 1 (2.7%) | 5 (11.1%) |
| yes | women | 157 (92.9%) | | | |
| | men | 63 (88.7%) | | | |
| no | women | 12 (7.1%) | | | |
| | men | 8 (11.3%) | | | |
| **First pain episodes** | | | | | |
| 1 years ago | | 32 (14.3%) | 24 (15.2%) | 4 (10.8%) | 4 (8.9%) |
| 2–3 years ago | | 56 (25.0%) | 40 (25.3%) | 12 (32.4%) | 4 (8.9%) |
| 4–6 years ago | | 53 (23.7%) | 29 (18.4%) | 10 (27.0%) | 14 (31.1%) |
| 7–9 years ago | | 31 (13.8%) | 20 (12.7%) | 6 (16.2%) | 5 (11.0%) |
| 10 and more | | 52 (23.2%) | 34 (21.5%) | 4 (10.8%) | 14 (31.1%) |
| **Number of pain episodes** | | | | | |
| 0 | | 22 (9.2%) | 16 (10.0%) | 2 (5.4%) | 4 (8.9%) |
| 1–5 | | 115 (50.2%) | 78 (49.4%) | 23 (62.2%) | 14 (31.1%) |
| 6–10 | | 46 (20.1%) | 26 (16.5%) | 4 (10.8%) | 16 (35.6%) |
| 11 and more | | 57 (24.9%) | 38 (24.1%) | 8 (21.6%) | 11 (24.4%) |
| **Cause of pain** | | | | | |
| lifting | | 140 (62.2%) | 91 (57.6%) | 19 (51.4%) | 30 (66.7%) |
| rotation of the torso | | 66 (29.3%) | 48 (30.4%) | 10 (27.0%) | 8 (17.8%) |
| bending over | | 84 (37.3%) | 58 (36.7%) | 13 (35.1%) | 13 (28.9%) |
| hypertrophy of the torso | | 22 (9.8%) | 14 (8.9%) | 5 (13.5%) | 3 (6.7%) |
| pushing weight | | 12 (5.3%) | 10 (6.3%) | 2 (5.4%) | 0 (0.0%) |
| pulling weight | | 47 (20.9%) | 29 (18.4%) | 5 (13.5%) | 13 (28.9%) |
| elusive cause | | 43 (19.1%) | 31 (19.6%) | 5 (13.5%) | 7 (15.6%) |
| **Activities and positions[a]** | | | | | |
| standing | | 90 (40.2%) | 56 (35.4%) | 16 (43.2%) | 18 (40.0%) |
| lifting | | 128 (57.1%) | 81 (51.3%) | 17 (45.9%) | 30 (66.7%) |
| bending over | | 77 (34.4%) | 51 (32.3%) | 14 (37.8%) | 12 (26.7%) |
| sitting | | 82 (36.6%) | 54 (34.2%) | 11 (29.7%) | 17 (37.8%) |
| rotation of the torso | | 51 (22.8%) | 38 (24.1%) | 9 (24.3%) | 4 (10.8%) |
| hypertrophy of the torso | | 21 (9.4%) | 12 (7.6%) | 7 (18.9%) | 2 (5.4%) |
| pulling weight | | 55 (24.6%) | 32 (20.3%) | 6 (16.2%) | 17 (37.8%) |

**Notes.**
[a] Activities and positions that intensify pain.

the vertical scale 0 corresponds to no limitation of motor activity, and 1 corresponds to partial limitation.

The analysis of data obtained from the NDI questionnaire revealed that in the studied group 151 (62.9%) PTs had no disability related to cervical pain, 66 (27.5%) had minimal disability, 21 (8.8%) had moderate disability, and 2 (0.8%) had severe disability. The results indicated minimal disability due to cervical pain.

The analysis of data obtained from the ODI questionnaire showed that 128 (53.3%) of PTs had no disability due to lumbosacral pain, 100 (41.7%) had minimal disability,

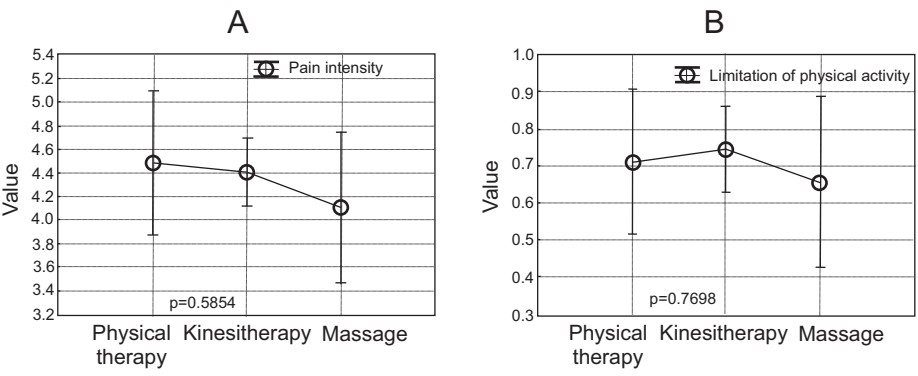

**Figure 3** Mean and 95.00% Confidence Intervals: (A) Last pain intensity (VAS); (B) limitation of physical activity.

**Table 3   Results of NDI and ODI in the study group and by speciality.**

| | NDI (n = 161) mean (SD); med; [min; max] | | | ODI (n = 196) mean (SD); med; [min; max] | | |
|---|---|---|---|---|---|---|
| All groups (N = 240) | 8.1 (6.3); 7; [0; 26] | Men (n = 33) | 5.7 (6.0); 3; [0; 24] | 6.0 (4.9); 5; [0; 25] | Men (n = 60) | 5.0 (4.9); 4; [0; 25] |
| | | Woman (n = 128) | 8.8 (6.2); 8; [0; 26] | | Woman (n = 136) | 6.4 (4.8); 6; [0; 22] |
| Kinesitherapy (n = 158) | 7.7 (5.5); 7; [0; 24] | Men (n = 19) | 6.2 (6.4); 3.5; [0; 24] | 5.6 (4.2); 5; [0; 25] | Men (n = 41) | 5.1 (4.6); 4; [0; 25] |
| | | Woman (n = 90) | 7.6 (5.6) 7; [0; 24] | | Woman (n = 87) | 5.8 (4.1); 5; [0; 18] |
| Physical therapy (n = 37) | 10.3 (7.9); 9.5; [2; 26] | Men (n = 2) | 3.5 (2.1); 3.5; [2; 5] | 7.4 (5.2); 6; [0; 17] | Men (n = 4) | 5.5 (2.9); 5.5; [2; 9] |
| | | Woman (n = 18) | 11.1 (7.9) 11.5; [2; 26] | | Woman (n = 29) | 7.7 (5.5); 8; [0; 17] |
| Massage (n = 45) | 8.1 (7.1); 6; [0; 23] | Men (n = 12) | 5.3 (5.9); 3; [0; 17] | 5.9 (6.4); 5; [0; 22] | Men (n = 15) | 4.7 (6.4); 3; [0; 22] |
| | | Woman (n = 20) | 9.8 (7.4) 8.5; [0; 23] | | Woman (n = 20) | 6.8 (6.3); 6; [0; 22] |

**Notes.**
NDI, Neck Disability Index; ODI, Oswestry Low Back Pain Disability Index.

11 (4.6%) had moderate disability, and one (0.4%) person had severe disability. Detailed statistics are presented in Table 3.

The Mann–Whitney U test was used to determine the difference in mean values of NDI and ODI between women and men. Based on the calculated test probability of NDI ($p = 0.002$) and ODI ($p = 0.024$), the null hypothesis was rejected, which means that the differences in the mean values of NDI and ODI values between men and women were significant. ANOVA showed that the average difference (median) between men and women in NDI in all groups of PTs was 5, whereas in the case of ODI it was 2. The differences in mean values by sex are presented in Fig. 4A. For women, NDI values were significantly

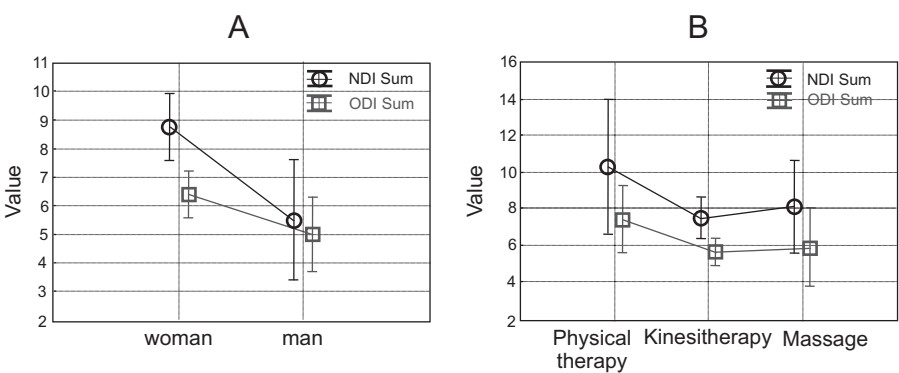

**Figure 4 NDI and ODI: woman and man (A) physical therapy, kinesitherapy, massage (B).**

higher than for ODI. NDI and ODI values for men were similar. Figure 4B shows a higher mean value for cervical pain in each group than for lumbar pain.

The Mann–Whitney U test was also used to determine the differences in mean values of NDI and ODI between specialities. The obtained values of $p = 0.171$ for NDI and $p = 0.074$ for ODI indicated the validity of the tested zero hypothesis, which means that the differences in NDI and ODI were not significant between specialities. Results of ANOVA showed that the mean difference in NDI between kinesitherapy and physical therapy was 1.5, and between kinesitherapy and massage 1.

Another analysis was performed (U M-W) to determine the nature of pain and the frequency of its occurrence in individual groups. The calculated value $p = 0.995$ indicated the validity of the tested zero hypothesis, which means that differences in pain between specialities were insignificant. The nature and frequency of pain were the same in kinesitherapists and physical therapists, which means that on average they had one episode of pain per year. Massage therapists experienced pain more often, once a month on average. The test for restriction of physical activity in individual groups ($p = 0.770$) showed no significant differences, and on average respondents had partially restricted activity due to pain.

Figure 5 presents the results of NDI and ODI normality tests in individual specialities. Since at least one of the selected tests invalidated the zero hypothesis, the hypothesis about the normality of data distribution was rejected in all cases. The figures show only results of the Shapiro–Wilk test. Considering ODI and the speciality of physical therapy, this test validated the zero hypothesis about the normality of distribution. However, one of the remaining tests rejected the zero hypotheses.

Using the NW Chi-square test (highest reliability) ($p = 0.285$) and Pearson's Chi-square test ($p = 0.215$) at the adopted level of significance (alpha = 0.05), a significant relationship between work experience (years) and the nature of pain and its frequency was found for all PTs. The results indicated that several episodes of pain were reported most often in PTs with 10-12 years of work experience.

The next step was to determine the correlation between NDI and ODI values, and individual variables at $p < 0.05$. The results obtained for ODI vs. NDI variables indicated

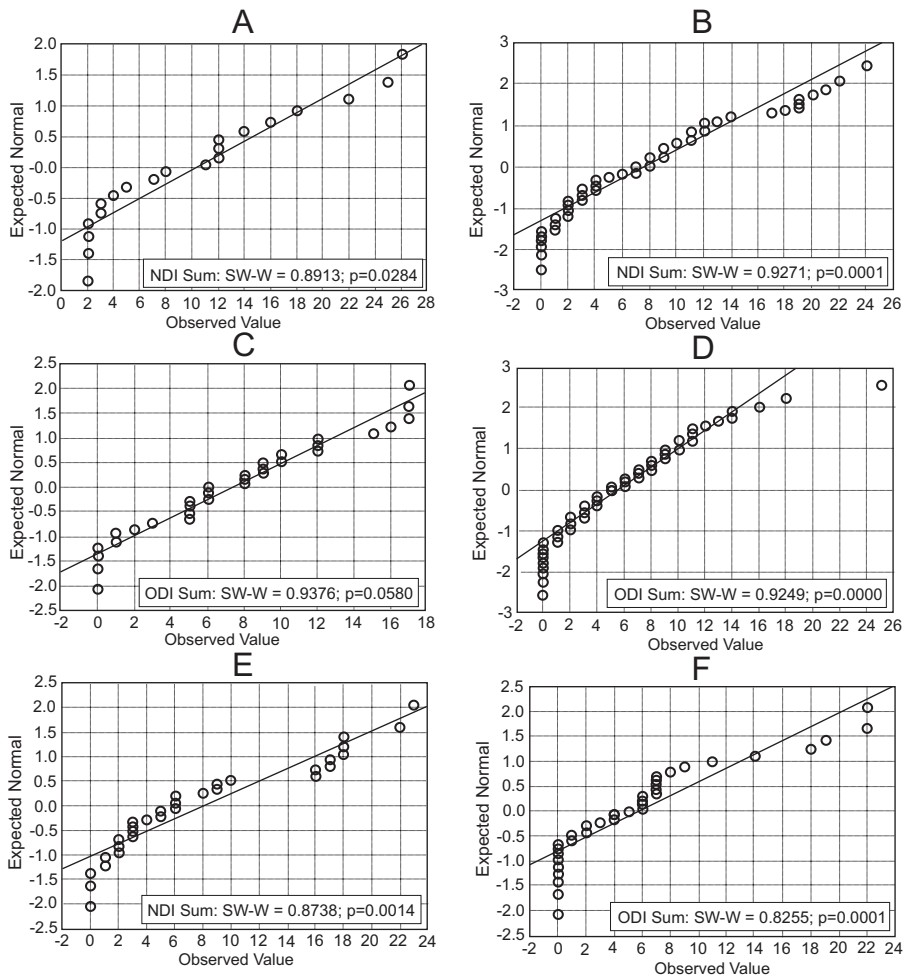

**Figure 5   NDI and ODI plot normality (A) NDI Physical therapy. (B) NDI Kinesitherapy. (C) ODI Physical Therapy. (D) ODI Kinesitherapy. (E) NDI Massage. (F) ODI Massage.**

a strong positive correlation $r = 0.8532$. There was a weak correlation between work experience and NDI ($r = 0.2700$), and ODI ($r = 0.2298$). There was practically no correlation between NDI ($r = -0.0013$) and ODI ($r = 0.0600$) and age of PTs. No correlation was found between NDI ($r = -0.0013$), ODI ($r = 0.0600$) and BMI. A moderate correlation was found between NDI ($r = 0.4369$) and ODI ($r = 0.5609$), and the use of painkillers. No correlation was found between work experience and duration of intensive and moderate physical activity.

The analysis of data concerning PTs' physical activity revealed that the majority of the respondents (76.7%) declared that during the last 7 days before the study they engaged in vigorous physical exercise causing rapid breathing and heartbeat for at least 10 min. These activities included, for example, aerobics, fast cycling or fast running. Kinesitherapists and massage therapists declared doing exercise twice on average, and physical therapists once a week. Only 17.9% of all respondents indicated that it was 4 times or more per week. In the case of physical therapists and kinesitherapists, the mean time of vigorous exercise

**Table 4** Effects of intensive and moderate levels of physical activity on the nature of pain and frequency of pain episodes in individual groups (NW Chi-square (highest reliability) and Pearson's Chi-square test).

| | Vigorous physical activity | | | |
|---|---|---|---|---|
| | Frequency | | Time | |
| | NW Chi-square | Pearson Chi-square | NW Chi-square | Pearson Chi-square |
| Kinesitherapy | p = 0.018 | p = 0.017 | p = 0.080 | p = 0.080 |
| Physical therapy | p = 0.066 | p = 0.067 | p = 0.010 | p = 0.027 |
| Massage | p = 0.433 | p = 0.571 | p = 0.205 | p = 0.414 |
| All groups | p = 0.037 | p = 0.029 | p = 0.003 | p = 0.003 |
| | Moderate level of physical activity | | | |
| Kinesitherapy | p = 0.021 | p = 0.011 | p = 0.183 | p = 0.312 |
| Physical therapy | p = 0.206 | p = 0.294 | p = 0.076 | p = 0.105 |
| Massage | p = 0.253 | p = 0.235 | p = 0.275 | p = 0.467 |
| All groups | p = 0.074 | p = 0.176 | p = 0.153 | p = 0.182 |

was defined as 30–50 min, while masseurs declared 10–30 min. There was no significant difference between men and women in the amount of exercise per week, while women declared a mean time of exercise of 10–30 min, while men declared 30–50 min.

Moderate/average levels of physical activity, i.e., movement causing a little faster breathing and a faster heartbeat (cycling, playing volleyball, very fast walking), were declared on average as 2 times a week in all groups. The declared mean duration of moderate physical activity was 30–50 min in masseurs, and only 10–30 min in the group of physical therapists and kinesitherapists. Only 23.3% of all respondents indicated that it was 4 times or more per week. Men preferred a longer duration of moderate physical activity (30–50 min) compared to women (10–30 min).

The analysis demonstrated that intensive and moderate levels of physical activity 3 times and more per week in the group of physical therapists and massage therapists reduced the frequency of pain episodes (up to a maximum of one per year). Lack of physical activity was associated with an increased number of pain episodes. The mean duration of intensive physical activity from 30 to 50 min in the group of masseurs significantly reduced the incidence of pain in this group. The mean exercise time of 30 to 50 min reduced the frequency of pain in all examined groups. In the other groups, the frequency and duration of intensive and moderate levels of physical activity did not affect the nature and frequency of pain episodes (Table 4).

# DISCUSSION

The results indicate that the pain described by the respondents has a complex aetiology, and several static and dynamic factors contributed to its onset. The analysis showed that NP and/or LBP was reported in all groups of physiotherapists (91.7%). The pain was more frequent in the lumbosacral spine (82%) than in the cervical spine (67%). However, the intensity of pain was higher in the cervical spine.

Our findings are similar to those of other studies carried out in various countries, such as Canada, the USA (*Vieira et al., 2016*; *Bork et al., 1996*; *Campo et al., 2008b*), Kuwait (*Shehab et al., 2003*), the United Kingdom (*Scholey & Hair, 1989a*), Slovenia (*Rugelj, 2003*), Turkey (*Molumphy et al., 1985*), and India (*Iqbal & Alghadir, 2015*). In the USA, the most common work-related locomotor disorder was LBP, which represents 45% (*Bork et al., 1996*). The lifetime prevalence of work-related pain in Kuwait was 70% (*Shehab et al., 2003*). In Canada, 49% of PTs reported work-related back pain (*Vieira et al., 2016*; *Campo et al., 2008b*). In India, as many as 92% of PTs indicated work-related musculoskeletal disorders, where 51% concerned LBP and 17% NP (*Iqbal & Alghadir, 2015*).

The frequency of musculoskeletal disorders associated with PTs' occupation is high. The resulting pain-related ailments significantly affect everyday activities and sometimes even force the PT to change their profession (*Iqbal & Alghadir, 2015*; *Mierzejewski & Kumar, 1997*). Mierzejewski and Kumar (*Mierzejewski & Kumar, 1997*) demonstrated that more than half (55.4%) of respondents with work-related LBP showed little or no disability. In our study, we observed that as many as 41.7% of PTs with LBP have minimal disability, while among those with NP 27.5% had minimal disability and 8.8% had moderate disability.

Activities such as lifting, tilting and twisting of the abdomen were indicated as the main cause of the first pain episode. Other authors indicated similar results (*Campo et al., 2008b*; *Scholey & Hair, 1989b*). The pain was increased by weightlifting, standing and sitting position, and bending of the abdomen. Unfortunately, PTs' work with the patient mostly requires the above-mentioned postures.

Considering the specific nature of PTs' work in various specialities, we have observed that the incidence of pain from the beginning of their career in kinesitherapists (91.1%), physical therapists (97.3%) and masseurs (88.9%) was at similar levels. *Vieira et al. (2016)* presented another stratification of the study group and reported that the most affected part of the body was the lower part of the back in PTs specializing in emergency care, geriatrics and paediatrics, and the neck in PTs specializing in orthopaedics and neurology. As far as workplaces are concerned, the lower back was most often affected in PTs working in specialized nursing homes, clinics and hospitals, and the neck in PTs working in academic and home environments.

Our study revealed a significant protective effect of regular exercise on the development of spinal pain. Similar results were observed in previous studies. The lack of regular exercise results in poor or no back support and improper body mechanics (*Glowinski & Krzyzynski, 2013*; *Terzi & Altın, 2015*).

A recently published meta-analysis suggests that a moderate to high level of physical activity in leisure time is associated with an 11–16% reduction in the incidence of episodic or chronic lower back pain (*Shiri & Falah-Hassani, 2017*). The explanations underlying the protective effects of exercise against chronic lower back pain (LBP) are unclear. Physical exercise in LBP may work by improving posture and muscle activation. However, there is no evidence linking the effects of exercise in LBP with changes in the musculoskeletal system (*Halliday et al., 2016*). There is solid evidence that LBP is best understood from a biopsychosocial perspective, as it may involve a combination of psychological, social, lifestyle and physical factors (*Kamper et al., 2015*).

## CONCLUSIONS

In the article we have shown that the profession of a physiotherapist in all specialties is associated with a high risk of pain in the cervical and lumbar spine.

In the entire study group, the incidence of pain in the lumbar spine was higher than that of the pain in the cervical spine. In the entire group of physiotherapists, longer work experience had a significant impact on the occurrence of back pain. There were no significant differences in the intensity of pain sensations between specialties. It has been shown, however, that due to the nature of their work, massage is exposed to a higher frequency of back pain compared to kinesitherapy and physical therapy. This may be due to the lack of compliance with ergonomic standards in the workplace.

In terms of activities causing and increasing the intensity of back pain in all specialties, lifting heavy objects/patients was the most frequently mentioned. Standing, sitting, and bending were listed as subsequent items that increased the occurrence of pain in all specialties. The implementation of preventive measures in the work of these specialists should be considered by teaching them behavioral patterns regarding correct posture and the use of special auxiliary devices. This can be achieved through the regular compulsory training of staff in the use of ancillary equipment and the enforcement of ergonomic standards in the workplace. It also seems reasonable to support the ergonomic aspects of the daily work of physiotherapists. Designers of the exoskeleton supporting the physiotherapist at work should take into account the stresses and pains affecting mainly the cervical and lumbar spine.

The level of disability caused by pain in the cervical or lumbar spine did not differ between specialties. In the entire study group of physiotherapists, the majority did not have disabilities caused by pain in the cervical (62.9%) and lumbar (53.3%) spine parts, but a significant percentage had a mild disability in the cervical (27.5%) and lumbar parts (41.7%) of the spine.

In the entire study group, the frequency and duration of physical activity contributed to a lower number of back pain symptoms. In individual specialties, it has been observed that the greatest benefits in the form of a smaller number of pain incidents from physical activity occur among massage, then among physical therapy, and the lowest among kinesitherapy. Employers should consider supporting physical therapists to improve their health, including organizing physical activity at the workplace as part of physioprophylaxis or preventive rehabilitation. The introduction of exercises aimed at restoring proper muscle tone and behavior minimizing the risk of overload caused by every day and professional activity will certainly lead to a reduction in absenteeism, and thus to an increase in the economic profit of employers.

The limitation of this study is the fact that women and men were not equally represented in the study group. The reason may be that the occupation of PT in Poland is preferred by women.

### Funding
The authors received no funding for this work.

### Competing Interests
The authors declare there are no competing interests.

### Author Contributions
- Sebastian Glowinski conceived and designed the experiments, performed the experiments, analyzed the data, prepared figures and/or tables, authored or reviewed drafts of the paper, and approved the final draft.
- Aleksandra Bryndal conceived and designed the experiments, performed the experiments, authored or reviewed drafts of the paper, and approved the final draft.
- Agnieszka Grochulska conceived and designed the experiments, performed the experiments, analyzed the data, prepared figures and/or tables, and approved the final draft.

### Human Ethics
The following information was supplied relating to ethical approvals (i.e., approving body and any reference numbers):

The study protocol was approved by the Bioethics Committee at the district medical chamber in Gdansk (KB-14/20).

### Data Availability
The raw data are available in the Supplemental File.

### Supplemental Information
Supplemental information for this article can be found online at http://dx.doi.org/10.7717/peerj.11715#supplemental-information.

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
