# Peer review of "Prevalence and risk of spinal pain among physiotherapists in Poland"

_PeerJ, doi:10.7717/peerj.11715_

## Round 0.1 · original submission · Major Revisions

The reviewers have made some important points that must be addressed. Specifically, the authors must make the contribution and importance of the work, as well as the methods, clearer.

Reviewer 1 ·

Basic reporting

no comment

Experimental design

no comment

Validity of the findings

no comment

Additional comments

I am so appreciated the author presenting this clinical useful research article emphasizing that high prevalence of spinal pain in physiotherapists. However, the results were well known. My comments are as follows
1.Please clarify the workplace of the patients were enrolled, in clinic of local hospital , or medical center.
2.The health professionals could self-treat pain using drug. Any healthy effects in your series.
3. Table 1, 2 : Any difference among the Kinesitherapy, physical therapy, Massage group?, please clarify
4. Please provide the number of daily treated patients in each groups?
5.The conclusion is nice

Reviewer 2 ·

Basic reporting

The introduction section is too lengthy, it will be better if it's made more concise and the total number of words is reduced.

Experimental design

1. How sample size was calculated?
2. The results section is also very extensive. If information is given in tables and figures then there is no need to write it again in the results section. If some information is not given in tables and figures then they can be written in the results section. Thus it would be better if the results section is also made more concise and the total number of words is reduced.
3. Instead of Chi2, Chi-square can be written.
4. Was any particular inclusion and exclusion criteria chosen? The participants studied had the age of a much wider range i.e. 19-63 years. Would it not be appropriate if a smaller range was chosen for study e.g. 20-40 years or 40-60 years so that the effects of old age and PT profession can be separated? Were PTs having traumatic spine injury or history of spinal surgery excluded from study?

Validity of the findings

1. Conclusion section is not written separately from discussion section.

Additional comments

1. The topic is relevant for PTs. But the overall manuscript should be made more concise and precise so that readers can get the best of your research without spending much time on reading or without spending much time searching for the best finding among all the findings.
2. What does W mean in line no. 127?

Reviewer 3 ·

Basic reporting

1. Revise the title and make it more specific to the regional population (country name) as so many researches on prevalence and risk have been published in different countries.
Consider writing country name at the last of this topic as “Prevalence and risk of spinal pain among physiotherapists in Poland!”
2. There is a need for Professional English editing by either an English native language editor/reviewer Or English language editing services agency as there is a lot of grammatical mistakes, repeated, spelling mistakes, the informal language used, and chaotic sentences found throughout the manuscript. English editing is strongly recommended.
3. What was the study design? Consider t write the study design in the methodology section of the Abstract and in the text of the manuscript as well.
4.Line 28: The first line should start mainly with addressing spinal pain as the most common problem among working professionals worldwide and are the common source of work-related disabilities and economic burden in the societies and so on… Then its prevalence and incidences, in general, indicating neck pain LBP.
5.Line 42: Avoid using non-formal words. Consider elaborating the term industrialized countries in a general term as “developing or developed countries”
6. Line 46: Consider writing “irregular or abnormal posture” instead of uncomfortable postures.
7. Line 51-55 & 56-59: Since occupation spinal neck pain and back pain are having the same root of origin, therefore, Both paragraphs should be merged into one paragraph and combine all risk factors and cause into 1-2 lines to describe the spinal pain including neck and back pain. Similarly, write 1-2 lines to explain the symptoms of NP & LBP. No need to describe in separate for NP and LBP.
8. Line 64-69: The work-related challenging factors including working environment and settings of equipment, the height of couches, warm-up exercises and muscle strength & conditioning for the physiotherapist, working hours, and rest-period between two consecutive patients, number of patients per day, etc. should be mentioned here. The mechanism of action of these conditions/postures/spinal malalignments, muscle weakness, etc., how badly affect the spinal alignment and make a reason for spinal neck and back pain.
9. Line 79: Write here the justification of the study that how this study would be new from other published studies that are similar so far? If it is specific to the country where research conducted, then add the name of that country in the title and here to justify this study as it was not done before here?

Experimental design

10. What was the study design?
Consider writing the study design in the methodology section of the Abstract and the text of the manuscript as well.
11. Line 94: Was it an actual or greater or lower sample size?
Explain in detail, How the sample size estimated? explain parameters, method of power analysis, and effect size and source of data to calculate the sample size!
12.Line 97: What was the inclusion and exclusion criteria for the study? Consider explaining the method and procedures for the sample collection and recruitment here in detail! How many samples meet the criteria and denied participation in the study? Add a paragraph:
13. Line 98-99: Was the questionnaire designed by the authors of this study tested for its accuracy, reliability & validity, if yes, then explain the procedure in brief?
14. Line 101-104: Was all sections of the questionnaire designed by the authors reviewed thoroughly and approved by the team of medical professionals such as senior consultant/physiotherapist, head/dean, or registered body of physiotherapy practice before being utilized? Did you find any weakness or drawback or extra strength of this questionnaire designed by the authors, if yes, consider explaining here?
15. Line 101-104: Consider writing the scoring method and psychometric analysis of the questionnaire designed by the authors here!
16. Line 110-112:Consider including the scoring method in detail and psychometric properties of the NDI questionnaire here!
17. Line 113-115: Consider including the psychometric properties of the ODI questionnaire here!
18. Line 118-112: Consider to explain in sentences rather than point-wise!
19. Line: 130: Here, all tests such as ANOVA, Tukey’s post hoc analysis, and others should be enumerated with their role to determine in the analysis!

Validity of the findings

20. Results: The authors have mentioned all values of quantitative data in the results section although they are already mentioned in the tables. Therefore, It is strongly recommended to revise the result writing and consider to write only those values which are not mentioned in the tables. No repetitions of data rather than refer to the tables will be encouraged!
21. Line 1323-136: Consider these lines to write as “there were 158 PTs specialized in kinesitherapy (65.8%), 37 physical therapists (15.4%), and 45 masseurs (18.8%) as presented in Table 1.”
22. Line:138: Consider to include standard deviation in the bracket along with mean as “the mean (SD) age of women …. And so on till the end of the paragraph.
Don’t explain the demographic details here if it is already present in tables. Explain those details which are not mentioned in the tables.
Don’t repeat every time like “mean age of women … mean height of women… mean weight of women… so on.” Consider writing in one sentence including all details.
23. Overall, the results section needs to be revised and limited to enumerate the data that are not mentioned in the table rather than the repetition of the same data. Authors should not write the result in a conclusive-form, they are required to just enumerate the findings in the result section.

Additional comments

Discussion:
24. Line:282: Give here an updated reference!
25. Line 291: A separate paragraph is required to write the mechanism of spinal pain (neck and back pain) due to lifting, sitting, and other mechanical/postural factors. How these factors create spinal pain and affect the activities of physiotherapists or other professionals?
26. Line 311-316 & 317-323: These paragraphs should be written in accordance with the purpose of the study. In actuality, It is the answer to your research hypothesis. Therefore, it should be written in a way to answer your hypothesis or fulfill the purpose and objective of your study. Consider to revise the whole paragraph and answer your hypothesis/purpose of the study.
27. Line:325: Give the reference to justify this statement?
28. Line 326: Taking the history of injuries is the basic part of the assessment. Your study was on prevalence and risk of spinal pain, then how you didn’t ask about the previous musculoskeletal injuries that might be imitating the occupational pain? Did you mention this point in the exclusion criteria of the study?

---

## Round 0.2 · Major Revisions

Most of the minor questions/comments/suggestions raised by the reviewers have been addressed, however, the authors did not fully reply to the major questions (the suitability, novelty, justification, self-made questionnaire validity, answering the hypothesis in the discussion, etc) that were raised by Reviewer 3.

Reviewer 1 ·

Basic reporting

no comment

Experimental design

no comment

Validity of the findings

no comment

Additional comments

I appreciate the author presenting this research article to emphasize the high prevalence and risk of spinal pain among physiotherapists in Poland. My comments are as follows
1. This manuscript is well designed but the presentation of results were not easy to read, for example (1) in Table 3 no painkiller used data but the authors mentioned in the manuscript (2). Figure 1, 2, 3 didn't show the p value.
2. Two questions need to be clarified
a. Previous study showed that pregnancy induces changes in spinal posture and weakens the joint
structure, which increases the risk for back pain. Any pregnancy in your cases?
b. Previous studies showed younger physiotherapists have a greater chance of developing back pain
because of they did not protect themselves in daily practice. Do you agree this mention?

Reviewer 2 ·

Basic reporting

no comment

Experimental design

no comment

Validity of the findings

no comment

Reviewer 3 ·

Basic reporting

The authors did not answer satisfactorily the following Line 61 and 79/questions no. 13-15, which was important to answer in the revised manuscript. [Line numbers are the same as in the original submitted manuscript ].

Line 61: The authors did not cover the suggested work-related challenging factors that contribute to spinal pain among physiotherapists. They must mention the complete risk factors as the reviewer suggested before.

Line 79: The authors did not add a paragraph for the justification of the study that how this study is novel from other previous studies as the reviewer suggested before. Without justification of the study, how the author can write their purpose. This comment must be addressed.

13. Line 98-99: Was the questionnaire designed by the authors of this study tested for its accuracy, reliability & validity, if yes, then explain the procedure in brief?
14. Line 101-104: Was all sections of the questionnaire designed by the authors reviewed thoroughly and approved by the team of medical professionals such as senior consultant/physiotherapist, head/dean, or registered body of physiotherapy practice before being utilized? Did you find any weakness or drawback or extra strength of this questionnaire designed by the authors, if yes, consider explaining here?
15. Line 101-104: Consider writing the scoring method and psychometric analysis of the questionnaire designed by the authors here!

Line 133-134. Why so many tests used for the normality of distribution. Either consider writing the use of each test Or just mention the one test which data you used in the table or study.

Experimental design

N/A

Validity of the findings

This question was not answered by the authors although it is a must answer.

In discussion, question and line numbers are the same as in the originally submitted manuscript.

26. Line 311-316 & 317-323: These paragraphs should be written in accordance with the purpose of the study. In actuality, It is the answer to your research hypothesis. Therefore, it should be written in a way to answer your hypothesis or fulfill the purpose and objective of your study. Consider revising the whole paragraph and answer your hypothesis/purpose of the study.

Additional comments

The authors of this study replied to most of the questions/comments raised by the reviewer however, there are few questions that are still required to be answered and they are very important for the suitability and acceptance of the manuscript. These questions are indicated with the previous question number and line numbers. Unless these questions will not be answered, the reviewer is not ready to accept this manuscript to be published.

---

## Round 0.3 · accepted · Accept

The authors have adequately addressed the reviewers concerns.